# Ambient Air Pollution and Acute Ischemic Stroke—Effect Modification by Atrial Fibrillation

**DOI:** 10.3390/jcm11185429

**Published:** 2022-09-15

**Authors:** Benjamin Yong-Qiang Tan, Jamie Sin Ying Ho, Andrew Fu-Wah Ho, Pin Pin Pek, Aloysius Sheng-Ting Leow, Yogeswari Raju, Ching-Hui Sia, Leonard Leong-Litt Yeo, Vijay Kumar Sharma, Marcus Eng-Hock Ong, Joel Aik, Huili Zheng

**Affiliations:** 1Department of Medicine, Yong Loo Lin School of Medicine, National University of Singapore, Singapore 117597, Singapore; 2Division of Neurology, Department of Medicine, National University Health System, Singapore 169857, Singapore; 3School of Clinical Medicine, University of Cambridge, Cambridge CB2 0QQ, UK; 4Department of Emergency Medicine, Singapore General Hospital, 1 Outram Rd., Singapore 169608, Singapore; 5Pre-Hospital and Emergency Research Centre, Health Services and Systems Research, Duke-NUS Medical School, Singapore 169857, Singapore; 6Environmental Monitoring and Modelling Division, Environmental Quality Monitoring Department, National Environment Agency, Singapore 228231, Singapore; 7Environmental Epidemiology and Toxicology Division, National Environment Agency, Singapore 228231, Singapore; 8National Registry of Diseases Office, Health Promotion Board, Singapore 168937, Singapore

**Keywords:** air pollution, ischemic stroke, atrial fibrillation, cerebrovascular accident, environmental epidemiology

## Abstract

Acute ischemic strokes (AIS) are closely linked with air pollution, and there is some evidence that traditional cardiovascular risk factors may alter the relationship between air pollution and strokes. We investigated the effect of atrial fibrillation (AF) on the association of AIS with air pollutants. This was a nationwide, population-based, case-only study that included all AIS treated in public healthcare institutions in Singapore from 2009 to 2018. Using multivariable logistic regression, adjusted for time-varying meteorological effects, we examined how AF modified the association between AIS and air pollutant exposure. A total of 51,673 episodes of AIS were included, with 10,722 (20.7%) having AF. The odds of AIS in patients with AF is higher than those without AF for every 1 µg/m^3^ increase in O_3_ concentration (adjusted OR [aOR]: 1.005, 95% CI 1.003–1.007) and every 1 mg/m^3^ increase in CO concentration (aOR: 1.193, 95% CI 1.050–1.356). However, the odds of AIS in patients with AF is lower than those without AF for every 1 µg/m^3^ increase in SO_2_ concentration (aOR: 0.993, 95% CI 0.990–0.997). Higher odds of AIS among AF patients as O_3_^−^ and CO concentrations increase are also observed in patients aged ≥65 years and non-smokers. The results suggest that AF plays an important role in exacerbating the risk of AIS as the levels of O_3_ and CO increase.

## 1. Introduction

Strokes are a major cause of mortality and morbidity worldwide, and around 30% of the global burden of strokes are thought to be attributable to air pollution [1]. In addition to modifiable behavioural and metabolic risk factors, environmental factors are the next largest contributors to stroke-related disability-adjusted life years. The effect of short-term and long-term air pollution exposure on cardiovascular diseases is being increasingly recognised. A meta-analysis of 98 studies across 28 countries in 2015 found that admission to hospital or mortality from acute ischemic stroke (AIS) was significantly associated with increasing concentrations of gaseous pollutants such as carbon monoxide (CO), sulphur dioxide (SO_2_), ozone (O_3_), and nitrogen dioxide (NO_2_), as well as particulate matter with diameter <2.5 µm (PM_2.5_) and <10 µm (PM_10_) [2]. There is early evidence that the susceptibility of traditional cardiovascular risk factors may alter the short-term risk of air-pollution-associated stroke. A previous study reports that PM_10_ increases the number of ischemic stroke events on the same day and the following day in patients with a history of strokes, but not in those without [3]. Furthermore, it is also reported that diabetes mellitus increases the susceptibility to AIS associated with PM_2.5_ exposure [4].

Atrial fibrillation (AF) is the most common arrhythmia of clinical significance, affecting 33 million individuals worldwide [5]. It is one of the biggest risk factors for AIS, especially in ageing populations. Although it is traditionally thought that fibrillation of the atrium causes stasis and thrombus formation, contemporary studies suggest a more complex mechanism between AF and AIS [6]. While environmental factors have significant impact on stroke risk, the interaction between AF and air pollution among AIS patients is unclear. A recent cohort study by Rhinehart et al. found that long-term exposure to PM_2.5_ increases the risk of AIS in patients with AF. On the contrary, long-term exposure to NO(x) shows no impact on the incidence of AF-associated AIS [7]. Hence, we aim to close this gap by investigating the effect of AF on the association of AIS episodes with short-term exposure to air pollutants (PM_2.5_, PM_10_, O_3_, NO_2_, SO_2,_ and CO).

## 2. Materials and Methods

### 2.1. Setting

Singapore is an urban city-state with air pollution levels comparable to other major cities, compliant with the United States Environmental Protection Agency standards and World Health Organization air quality guidelines [8]. It has a population density of 8358 per km^2^, and a tropical rainforest climate characterized by uniform warm temperatures, high humidity, and abundant rainfall. Its major sources of air pollution include the usual urban sources such as vehicle traffic and industrial activity, cargo ships crossing the Singapore Strait, and episodic transboundary pollution from wildfires and land-clearing activities in neighbouring Southeast Asian countries [8]. With wide day-to-day fluctuations in air quality, Singapore is well-placed to investigate the effect of air pollution on population health.

### 2.2. Study Population and Outcome Data

Daily cases of AIS in Singapore from 2009 and 2018 were collected from the Singapore Stroke Registry (SSR). The SSR receives case notifications from all public healthcare institutions, the Ministry of Health, and the national death registry. Data from multiple notification sources are linked using the unique National Registration Identity Card number to remove duplicate cases. The International Classification of Diseases (ICD) diagnosis codes used to identify stroke cases included ICD-9 codes 430 to 437 (excluding 432.1 and 435) prior to 2012, and ICD-10 codes I60 to I68 (excluding I62.0 and I62.1) from 2012 onward. All the cases captured by the SSR were diagnosed as strokes by a medical practitioner, defined as the presence of neurological deficit lasting more than 24 h, supported by appropriate neuroimaging and vessel imaging. The registry coordinators confirmed the diagnosis and etiology of stroke by assessing patients’ medical records before extracting detailed individual-level clinical data. Although the SSR captures AIS and haemorrhagic stroke, haemorrhagic strokes were excluded in this study due to differences in the effect of air pollution on AIS and haemorrhagic strokes [2,9]. AIS that developed while the patient was hospitalised for other condition were excluded, due to the likelihood of a different stroke pathophysiology, such as resultant from cardiovascular intervention and acute medical illness.

Data on the presence of AF (previous or recent diagnosis) and other clinical characteristics were extracted from the SSR. Medically, AF was defined as the absence of P waves and irregular RR intervals in the absence of atrioventricular conduction impairment on 12-lead ECG, or for ≥30 s on a single-lead ECG [10]. If AF was diagnosed by a registered medical practitioner and/or patient was treated for AF, as documented in the patient’s medical records, it is captured by the independent registry coordinators. Data from the SSR are subject to annual audit for accuracy and inter-rater reliability. Outlier and illogical data were reviewed by consensus among the registry coordinators.

### 2.3. Exposure and Meteorological Data

The concentrations of six air pollutants (PM_2.5_, PM_10_, O_3_, NO_2_, SO_2,_ and CO) from 2009 to 2018 were obtained from the National Environment Agency (NEA) Singapore. The NEA collects data on pollutant levels via 22 remote stations, and data on temperature, humidity, and rainfall via 11 weather stations across Singapore. The daily 24 h mean concentrations of PM_2.5_, PM_10_, O_3_, NO_2_, SO_2,_ and CO were calculated for each monitoring station and averaged across the available stations to obtain the pollutant concentrations in Singapore.

The individual-level data from the SSR were matched with the NEA data comprising daily pollutant concentrations using the hospital arrival date for strokes in SSR data and the daily date in NEA data. The eventual matched data for analysis were individual-level, assuming all patients who were admitted for strokes on the same day were exposed to the same level of pollutants on that day (based on the 24 h averages).

The Centralised Institutional Review Board and the Domain Specific Review Board granted ethics approval for this study with a waiver of patient consent as the data used in this study are anonymised (CIRB Ref: 2017/2380).

### 2.4. Statistical Analysis

Baseline characteristics of patients with AF and those without were compared using the chi-squared test for categorical variables and Mann–Whitney U test for continuous variables. The case-only approach was used to examine the effect of AF on the association between air pollutants and AIS. The case-only approach is widely used in gene–environment studies, and its use in time-series studies was first proposed by Armstrong [11]. Unlike genetic epidemiology, the key features of the case-only approach in the context of time-series studies are that the risk factors of interest (e.g., air pollution) vary in time but not over individuals at the same time, while the modifiers (e.g., AF) vary over individuals but not over time (or vary slowly over time). This orthogonality fulfils the assumption of independence of the exposure of interest and the potential modifier in the population from which cases arose [11]. Although the case-only approach was based on AIS patients only, its inference is akin to modelling the interactions in the conventional Poisson model applied to the entire population that contributed to AIS incidence.

Logistic regression models were fitted on the individual-level data of AIS episodes, in which the dependent variable was the presence or absence of AF, and the independent variables were the air pollutant concentration (PM_2.5_, PM_10_, O_3_, NO_2_, SO_2,_ or CO), mean temperature, relative humidity, and total rainfall on the day of arrival at the hospital. The adjustment for temperature, relative humidity, rainfall, and day of arrival were similar to previously published studies [2]. Odds ratio > 1 indicates that those with AF have a higher risk of AIS associated with air pollutant exposure than those without AF. Notably, odds ratio < 1 does not necessarily indicate a lower risk of AIS associated with air pollutant exposure. It could be that the increase in risk of AIS associated with air pollutant exposure is less pronounced for those with AF than those without AF.

All models included only single pollutant to avoid multi-collinearity among the pollutants. Empirically, as there was no seasonal variation of overall AIS incidence and AIS incidence in patients with AF in Singapore, seasonal effects were not included in the models. Additionally, the analyses of the effect modification by AF were stratified based on age at AIS (<65 years and ≥65 years) and smoking status (current/ex-smokers and non-smokers), to assess if AF remained as an effect modifier of the air pollutant–AIS relationship in pre-specified subgroups of AIS patients.

All statistical analyses were performed using STATA SE 13, and *p* values < 0.05 were considered statistically significant.

## 3. Results

### 3.1. Baseline Characteristics of Patients with AIS

In 2009 to 2018, 51,673 episodes of AIS were identified from the SSR, of which 10,722 (20.7%) have AF. Those with AF tend to be older, with a median age of 77 years (interquartile range, IQR 69–84 years), compared to 65 years (IQR 57–75 years) for those without AF (*p* < 0.001). The proportion of males is lower among patients with AF (45.6% versus 62.2% among those without AF, *p* < 0.001). Table 1 presents the baseline characteristics of the AIS patients by AF status.

### 3.2. Air Pollutant Concentrations and Meteorological Levels

Table 2 presents the descriptive summary of the air pollutants and meteorological factors in 2009 to 2018, while Figure 1 shows the trends of the daily average concentration of the air pollutants. PM_2.5_, PM_10_ and CO appear to have similar trends, with their highest concentrations recorded in June 2013 and October 2015, coinciding with periods of transboundary haze that Singapore experienced. NO_2_ and SO_2_ generally demonstrate similar trends, with at least one periodic within-year peak. O_3_ does not appear to exhibit a regular cyclical trend.

### 3.3. Effect Modification of AF on Air Pollutant and AIS

The odds of AIS associated with O_3_ and CO among patients with AF are higher than those without AF. After adjusting for daily mean temperature, relative humidity, and total rainfall, the adjusted odds ratio (aOR) is 1.005 (95% CI 1.003–1.007) with every 1 µg/m^3^ increase in O_3_ concentration (Figure 2A), and 1.193 (95% CI 1.050–1.356) with every 1 mg/m^3^ increase in CO concentration (Figure 2B) among patients with AF compared to those without AF. However, the odds of AIS associated with SO_2_ among patients with AF are lower than those without AF (aOR: 0.993, 95% CI 0.990–0.997, Figure 2C). The odds of AIS associated with PM_2.5_, PM_10,_ and NO_2_ among patients with and without AF are not significantly different (Appendix A).

### 3.4. Effect Modification of AF on Air Pollutants and AIS in Subgroups of AF Patients

Higher odds of AIS associated with O_3_ (aOR 1.006, 95% CI 1.003–1.008, Figure 2A) and CO (aOR 1.289, 95% CI 1.109–1.498, Figure 2B) among patients with AF and the lower odds of AIS associated with SO_2_ (aOR 0.994, 95% CI 0.990–0.999, Figure 2C) among patients with AF compared to those without AF are also observed in those aged ≥65 years, but not in those aged <65 years. However, the aOR between AF and non-AF patients in those aged ≥65 years does not differ significantly from the corresponding aOR in those aged <65 years for all the six pollutants (*p*-value of difference in aOR > 0.05).

Lower odds of AIS associated with SO_2_ (aOR 0.989, 95% CI 0.983–0.996, Figure 2C) among patients with AF compared to those without AF are also observed in those who smoke but not in those who do not smoke. Higher odds of AIS associated with O_3_ (aOR 1.006, 95% CI 1.003–1.009, Figure 2A) and CO (aOR 1.177, 95% CI 1.010–1.372, Figure 2B) among patients with AF compared to those without AF are also observed in those who do not smoke, but not in those who smoke. However, the aOR between AF and non-AF patients in those who smoke is not significantly different from the corresponding aOR in those who do not smoke for all the six pollutants (*p*-value of difference in aOR > 0.05). The modification effect of age group and smoking status on air pollutant and AIS is shown in Appendix A.

## 4. Discussion

Our study shows that, after adjustment for daily mean temperature, relative humidity, and total rainfall, AF increases the odds of AIS associated with O_3_ and CO exposure, but reduces the odds of AIS associated with SO_2_. Effect modification of AF is statistically significant in older adults aged ≥65 years and non-smokers, but not in those <65 years old and smokers. To the best of our knowledge, this is the first study to examine the effect of AF on the risk of AIS associated with short-term air pollutant concentration.

We found that AF significantly modifies the association of AIS with short-term air pollution, especially O_3_ and CO. Previous studies show that O_3_ concentration is significantly associated with paroxysmal AF detection [12] and recurrent ischemic cerebrovascular events, even by short-term exposures at low concentrations [13]. Same-day O_3_ concentration is also associated with ischemic stroke hospitalization among patients above the age of 65 years, but not haemorrhagic stroke [14]. Short-term exposure to CO was also previously established to be associated with ischemic stroke and AF separately [2,15], but evidence for interaction effects in those with AF and stroke is lacking. We found that in those with AF, the adjusted odds of AIS are higher than those without AF for each unit increase in O_3_ (aOR 1.005) and CO (aOR 1.193) concentration.

A previous cohort study found that at the highest quartile of long-term PM_2.5_ air pollution, measured by the average pollution level over a year, the risk of AIS associated with AF is 1.2 times of the lowest quartile [16]. However, that study only analysed PM_2.5_ in an area with heavy industrial activity and one of the top ten most heavily polluted cities in the US, which may not represent a typical urban city [16]. In contrast, we found that, in the short term, AF does not significantly increase the risk of AIS associated with PM_2.5_. Although the median PM_2.5_ is similar between this study and that by Rhinehart et al. [16], the peak PM_2.5_ may differ, and it remains possible that PM_2.5_ may exert an effect beyond a certain threshold. Although PM_2.5_ is linked to AIS caused by small-vessel occlusion and large-artery atherosclerosis [4], the association with cardioembolic strokes is controversial, and evidence is mixed. One study found that PM_2.5_ lowers the risk of cardioembolic stroke [4], while another study found that PM_10_ and SO_2_ are associated with increased risk [17]. As strokes due to AF are classed as cardioembolic, it may be possible that they are less susceptible to the effects of PM. Further research into the association of fine ambient particulate matter level, AF, and strokes is needed to better establish its effect on this vulnerable population.

Interestingly, this study found that patients with AF have a lower risk of AIS associated with short-term exposure of SO_2_ compared to those without AF. The impact of SO_2_ on AIS has been mixed. Some studies show that SO_2_ is associated with hospitalization for ischemic and haemorrhagic strokes and stroke mortality [18], while others do not [9]. SO_2_ is found to be associated with stroke mortality only in the winter but not summer in a study based in Wuhan, China [19]. In animal studies, SO_2_ elevates levels of interleukin-1b, tumour necrosis factor-a, and cyclooxygenase-2, which may contribute to a pro-inflammatory and pro-thrombotic environment, thus, encouraging thrombus formation and stroke. However, there are few data on the interaction between AF and SO_2_, and the mechanism underlying our finding is unclear.

Several mechanisms between air pollution and stroke have been proposed. Continued long-term exposure to air pollutants over months to years may impair endothelial function, and might activate the sympathetic nervous system, leading to vasoconstriction, increased blood pressure, and risk of thrombosis [20]. Particulate matter pollution may increase coagulability by activating alveolar inflammatory response and triggering release of inflammatory cytokines [2]. This may compound with the increased platelet activation, thrombin generation, and prothrombotic activation in AF, further increasing the risk of stroke [6]. AF also induces endothelial dysfunction and inflammation, and causes stasis of blood in the atrium [21]. The combination of air pollution and AF causing an inflammatory pro-thrombotic environment may explain the augmentation of risk of AIS due to O_3_ and CO found in this study. O_3_ is proposed to alter autonomic balance and induce cardiac arrhythmias, leading to mechanical and electrical changes in the heart [22]. Long-term CO exposure in the urban environment can also cause adverse cardiac effects by altering redox handling, ion homeostasis, intracellular calcium handling, and sympathovagal balance [23]. The cardiovascular effects of O_3_ and CO may further impact on AF. Although it is theoretically possible that anticoagulation in patients with AF may influence the interaction between AF, air pollution, and AIS, it would be expected that anticoagulation would reduce the risk of AIS [24], rather than increase AIS risk associated with O_3_ and CO. Furthermore, the prevalence of anticoagulation in the general AF population is only around 50% [25]. Additional research on the impact of air pollution on the various stroke subtypes, and its interaction with AF and anticoagulation, is needed to clarify the mechanisms behind the observed relationships in this study.

We found that older adults ≥65 years with AF are particularly vulnerable to AIS linked to increasing concentration of O_3_ and CO. A study performed in China also found that short-term O_3_ and PM_2.5_ exposure is significantly associated with strokes in those ≥40 years old, but not those aged <40 years [26]. Age is a clear risk factor for AIS in AF, and is included in the CHA_2_DS_2_VASc score, a commonly used stroke risk prediction score for non-valvular AF [27]. Older individuals may be more susceptible to air pollution due to the chronic inflammatory and pro-thrombotic state that accompanies aging, increasing the effects of pollutants on the cardiovascular system [28].

Our results suggest that AF increases the susceptibility of non-smokers to AIS associated with O_3_ and CO. Smoking-induced preconditioning, or the smoker’s paradox, is a controversial idea that smokers have a lower risk of mortality and morbidity after myocardial infarction, due to protective effects of reactive oxygen species on the endothelium against ischemic and reperfusion injury [29]. This may explain the finding of smokers having reduced vulnerability to air pollutants. However, other studies suggest that the protective effect of smoking is lost when adjusting for confounding factors [30], therefore it is possible that the apparent lower risk in smokers may be due to unaccounted variables.

We acknowledge some limitations of this study. The main limitation of this study is ecological fallacy, where exposure to air pollution at an individual level is inferred from the population level pollutant concentration. The concentration of pollutants used in this study is aggregated from data collected from multiple stations across Singapore, which may differ from the actual concentration individuals were exposed to. We were also unable to analyse the effect modification of AF on the prognosis of AIS patients exposed to air pollutants due to the lack of individual-level data on air pollution exposure. In this study, the event date of AIS is assumed to be the date of arrival to hospital, rather than the onset date of AIS, and thus, the recorded event date would be later than the actual onset of AIS in some cases. However, the advantage of using the date of arrival to hospital is that it is not dependent on the memory of the patient and, therefore, may be a more reliable and consistent measure. Due to the lack of anticoagulation data available, we were unable to assess the impact of pre-stroke anticoagulation on the joint influence of AF and air pollution on strokes. Instead of estimating the excess risk of AIS associated with air pollutants among patients with AF in absolute terms, the case-only approach estimated it relative to the risk in those without AF. Consequently, it is not possible to ascertain whether the negative association for AIS represents a decrease in risk or a less pronounced increase in risk among patients with AF compared to those without AF. However, the benefit of using the case-only approach in our study is that it eliminates the need of complex modelling of the change in risk of AIS over time as the result of factors other than air pollutants and meteorological factors. It also reduces the potential bias from model misspecification, as a variable must be associated with both the air pollutants and AIS for it to be a confounder. In this study, we analysed incidence of AIS associated with concentration of air pollutants on the same day of hospitalization, and further study is needed to explore the short-term and long-term lag effects of air pollution on AIS.

## 5. Conclusions

The odds of AIS associated with O_3_ and CO are higher in patients with AF than those without AF, suggesting that AF plays an important role in exacerbating the risk of AIS at increasing levels of O_3_ and CO. Effect modification of AF is stronger among older adults aged ≥65 years and non-smokers. Further studies are needed to better characterise the mechanism of this interaction, and to develop public health policies to reduce the health effects of air pollution on the urban population.

## Figures and Tables

**Figure 1 jcm-11-05429-f001:**
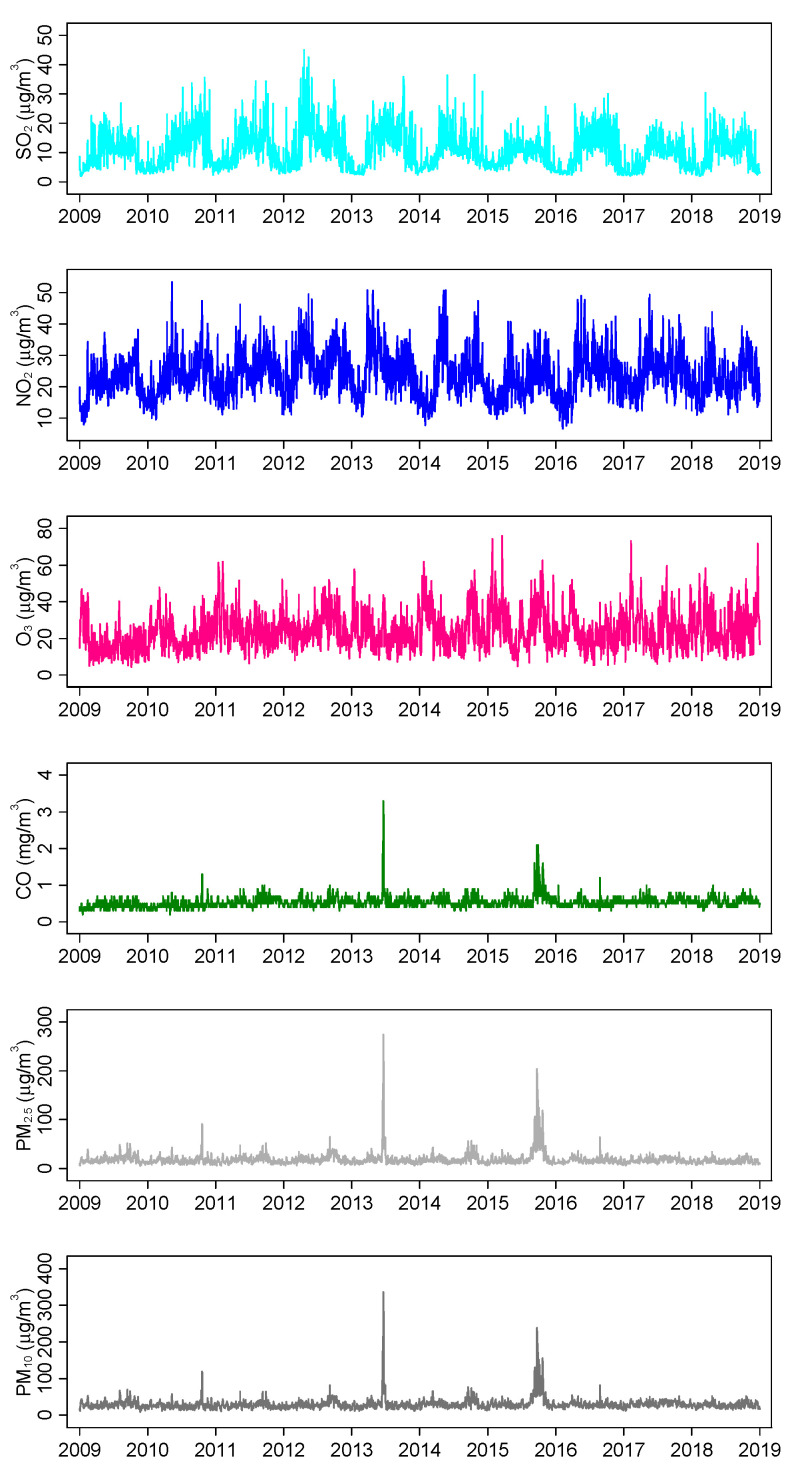
Trend of daily air pollutant concentrations in Singapore from 2009–2018.

**Figure 2 jcm-11-05429-f002:**
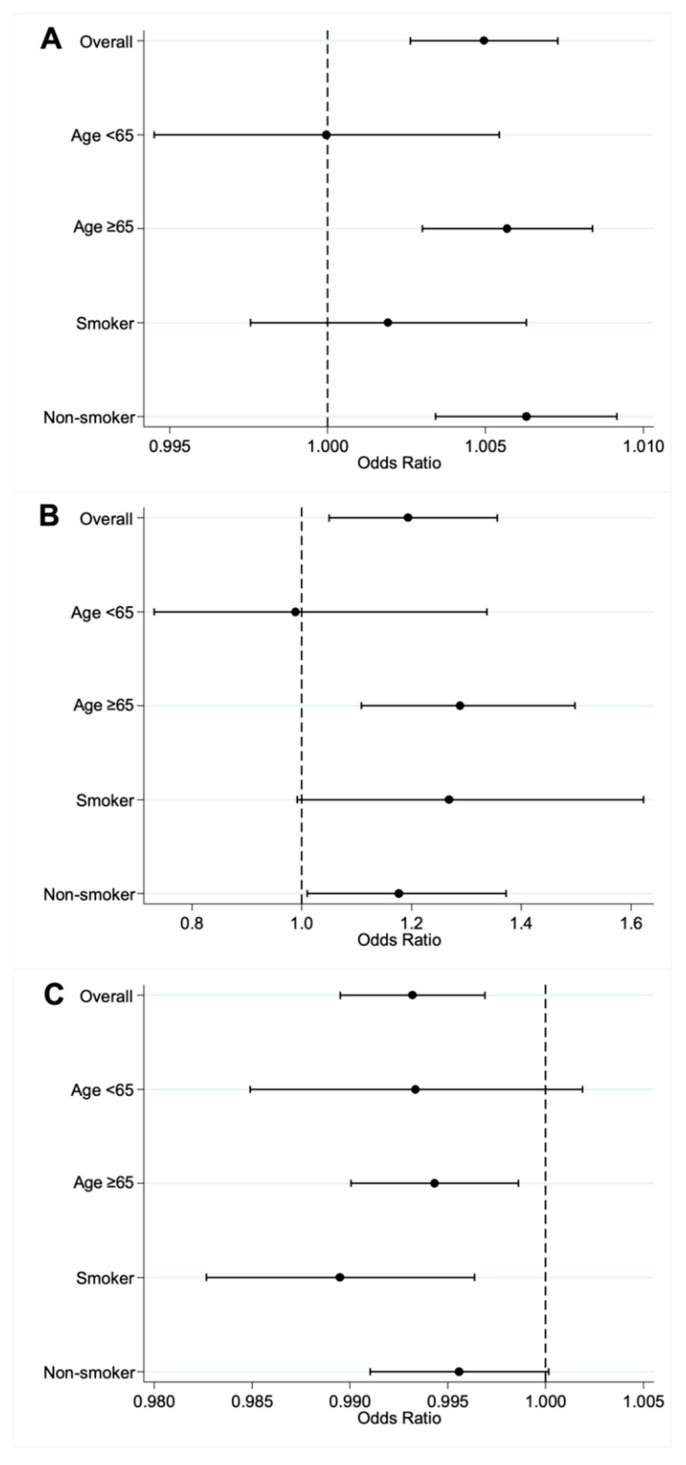
Adjusted odds ratio of AIS for (**A**) O_3_ exposure, (**B**) CO exposure, and (**C**) SO_2_ exposure among all and subgroups of patients with AF compared to those without AF.

**Table 1 jcm-11-05429-t001:** Characteristics of patients with AIS in 2009–2018 by atrial fibrillation status (*n* = 51,673 episodes).

	No AF *n* = 40,951	With AF *n* = 10,722	*p*-Value
Age, median (IQR)	65 (57–75)	77 (69–84)	<0.001
Male gender, *n* (%)	25,478 (62.2)	4888 (45.6)	<0.001
Ethnicity, *n* (%)			<0.001
Chinese	30,312 (74.0)	8627 (80.5)	
Malay	6664 (16.3)	1533 (14.3)	
Indian	3256 (7.9)	386 (3.6)	
Others	719 (1.8)	176 (1.6)	
Diabetes mellitus, *n* (%)	18,690 (45.6)	4424 (41.3)	<0.001
Hypertension, *n* (%)	33,116 (80.9)	9595 (89.5)	<0.001
Hyperlipidaemia, *n* (%)	37,363 (91.2)	9587 (89.4)	<0.001
Smoking status, *n* (%)			<0.001
Current smoker	11,471 (28.0)	1216 (11.3)	
Ex-smoker	5866 (14.3)	1687 (15.7)	
Non-smoker	22,617 (55.2)	7422 (69.2)	
Unknown smoking status	997 (2.4)	397 (3.7)	

Abbreviations: AF—atrial fibrillation; IQR—interquartile range.

**Table 2 jcm-11-05429-t002:** Descriptive summary of the meteorological factors and air pollutants in 2009–2018 (*n* = 3652 days).

	Mean (SD)	Median (IQR)	Minimum	Maximum
**Meteorological factors**				
Average temperature, °C	27.8 (1.1)	27.9 (27.1–28.7)	22.8	30.8
Relative humidity, %	79.5 (5.4)	79.4 (75.6–83.2)	59.2	96.9
Total rainfall, mm	5.5 (12.6)	0.0 (0.0–4.2)	0.0	216.2
**Air pollutants**				
PM_2.5_, µg/m^3^	18.4 (12.9)	15.9 (12.7–20.5)	5.1	274.4
PM_10_, µg/m^3^	30.0 (15.8)	27.3 (22.7–33.4)	9.7	335.9
O_3_, µg/m^3^	24.3 (10.1)	22.5 (17.3–29.8)	4.4	76.0
NO_2_, µg/m^3^	23.9 (7.0)	23.3 (18.8–28.4)	6.8	53.4
SO_2_, µg/m^3^	10.8 (6.1)	10.2 (5.6–14.4)	2.0	45.0
CO, mg/m^3^	0.5 (0.2)	0.5 (0.5–0.6)	0.2	3.4

Abbreviations: CO—carbon monoxide; IQR—interquartile range; NO_2_—nitrogen dioxide; O_3_—ozone; PM_2.5_—particulate matter 2.5; PM_10_—particulate matter 10; SO_2_—sulphur dioxide.

## Data Availability

The data that support the findings of this study are available from National Registry of Diseases Office and National Environment Agency Singapore upon reasonable request.

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
