# Peer review of "Ambient Air Pollution and Acute Ischemic Stroke—Effect Modification by Atrial Fibrillation"

_jcm, 2022, doi:10.3390/jcm11185429_

Round 1
Reviewer 1 Report
Dear authors, your research is of greater interest for stroke doctors but I have 1 question and 1 suggestion before publication:- are you sure that you correctly used the case only statistical method? in fact, in my mind, this method is valid only when variables are not dependent between each other. But, in this case,AF, air pollution and AIS are notoriously correlated. Can you clarify this point?
- do you have data about anticoagulant treatment in your database? It would be interesting to add this variable to your analysis in order to better assess the joint influence of AF and pollution on the stroke.
Thank you
Reviewer 2 Report
This article is well written in English, but I have 3 questions:
1. Generally, if we analyze the effect modification of AF, the analysis should be based on the whole population including patients with AF and without AF, and non-patients with AF and without AF. However, findings from the current study only indicate the risk of AIS patients with AF exposed to air pollutants, not the effect modification of AF on the risk of AIS exposed to air pollutants.
I suggest you can analyze the effect modification of AF on the prognosis of AIS patients exposed to air pollutants using the present data.
2. Although, empirically, there is no seasonal variation of AIS incidence in Singapore, data on whether there is a seasonal variation of AIS patients with AF should be presented.
3.Air pollutants have a lag effect on disease. When analyzing the impact of air pollutants on disease, the short-term lag effect should be considered.
Reviewer 3 Report
The authors examined the effect of atrial fibrillation (AF) on the association of acute ischemic stroke (AIS) episodes with short-term exposure to six criteria air pollutants (PM2.5, PM10, O3, NO2, SO2 and CO). This research is interesting, but the major concerns, especially on result explanation and interpretation, must be intensively improved.
Review comments
Major comments
1. Abstract is very confusing. Specifically, the authors stated that the odds of atrial fibrillation (AF) increased for every 1 µg/m3 increase in O3 and 1mg/m3 increase in CO, suggesting that authors have explored the association between air pollutants and AF (i.e., not and acute ischemic stroke: AIS), which is not aligned the title of manuscript. This result should be interpreted, for example, as the odds of AIS in patients with AF was higher than those without AF with OR of 1.005 (95% CI 1.003-1.007) per 1 µg/m3 increase in O3. Isn’t it? If so, please correct the sentences. Moreover, the authors stated that odds of AF declined in AIS patients for every 1 µg/m3 increase in SO2 concentration. This sentence obviously means the effect modification of AIS on the association between AF and SO2, which is different from the manuscript’s title. In this matter, authors must state in the other way around by showing the odds of AIS per unit increase in SO2 concentration in AF patients. Please carefully check your analyses and results.
2. The authors calculated 24-hr average concentration of all six air pollutants as proxy of exposure. However, daily 8-hr maximum concentrations of O3 and CO have been considered most of the time in prior studies. Moreover, 8-hr average concentration is also set as Singapore Ambient Air Quality Targets. Why did authors not consider 8-hr maximum for O3 and CO in this study?
3. I understand that the unit of analysis for air pollutant was in daily basis (n = 3652 days in Table 2). However, that of AIS is individual data (n = 51,673 episodes in Table 1), am I right? If so, how did you match air pollution data, as well as meteorological data, to every single AIS case?
4. Lines 183-190 (page 9), the explanation of effect modification of AF on the association between air pollutants and AIS is very confusing, please rewrite those sentence. I suggest as “The odds of AIS associated with O3 and CO among patients with AF were higher than those without AF with adjusted OR of 1.005 (95% CI: 1.003-1.007) per 1 µg/m3 increase in O3 concentration (Figure 2A) and 1.193 (95% CI: 1.050-1.356) per 1 mg/m3 increase in CO concentration (Figure 2B) after adjustment for daily mean temperature, relative humidity and total rainfall. However, the odds of AIS associated with SO2 among patients with AF were lower compared to those without AF with adjusted OR of 0.993 (95% CI: 0.990-0.997 (Figure 2C)”. Please carefully check your writing and this interpretation must be consistent across the manuscript. Remember that you explored the effects of air pollutants on AIS among AF status. Therefore, OR of AIS should be reported, and it should not be OR of AF (Please rewrite this throughout the manuscript).
5. Lines 183-190 (page 9), the authors stated that “the odds of AF were not affected by PM2.5, PM10 and NO2 concentrations (Supplementary Material)”. Again, this should be the odds of AIS, not the odds of AF. Moreover, this interpretation is generally not appropriate, although the OR was not significant, it does not mean it is not affected. Interpretation as “the odds of AIS associated with exposure to PM2.5, PM10 and NO2 among patients with and without AF was not significantly different” may be more understandable. Please consider about this.
6. Result explanation and interpretation is major issue of this manuscript, please reconsider in every single heading throughout the manuscript.
7. Line 213 (page 12), the authors justified whether the modified effect of AF on air pollutants–AIS association in different subgroups was statistically significant by considering the overlapping 95% CI, but it is not sufficiently clear. Hence, estimating p-value of the difference is preferable. Please refer to published article, such as https://doi.org/10.1016/j.scitotenv.2020.136985 (Bergmann et al., 2020)
8. Discussion seems to be inconsistent with what you found in this study. Specifically, you did not observe the significant effect modification of AF on the association between particulate matter (both PM2.5 and PM10) and AIS, but you discussed as like you observed the significance. In this matter, you should discuss why significant effect modification of AF on the association between particulate matter and AIS is not observed, compared to the similar previous studies.
9. Lines 229-231 (page 15), it is very confusing, where authors indicated the effects of PM2.5 on stroke incidence in smoking and non-smoking groups of previous studies, but indicated those of PM10, SO2, and CO of current study. Is it comparable or informative? Please reconsider carefully.
Minor comments
1. In Table 1, p-value from which statistical test should be noted.
2. In Figure 1, the unit of CO in Y-axis should be mg/m^3
3. In Table 3, the effect modification of smoking behavior on the association between PM10 and AIS seems to be weird. In particular, p-value for PM10 should be higher than 0.05 because 95% CI overlaps 1, which is not statistically significant and similar to that of PM2.5, O3, and NO2. Please check them carefully.
Round 2
Reviewer 2 Report
The revised version is very clear, no other comments
Reviewer 3 Report
Thank you for your effort addressed my concerns. It was amazing. I have no further comment.